# Mesoscopic insights into effects of electric field on pool boiling for leaky dielectric fluids
Geng Wang[1], Junyu Yang[2], Timan Lei[3], Linlin Fei[4], Xiao Zhao[1], Jianfu Zhao[1,5], Kai Li [1,5] ✉ & Kai H. Luo [3] ✉

The electric field is known as an effective approach to improving pool boiling. However, there has been limited research on electric field-enhanced boiling of leaky dielectric fluids and the associated bubble dynamics. In this work, we employ a mesoscopic multiphase lattice Boltzmann method to perform large-scale three-dimensional simulations of electric field-enhanced pool boiling in leaky dielectric fluids. Our findings confirm that, compared to conventional pool boiling, electric field-enhanced pool boiling significantly increases heat transfer efficiency in the transition boiling regime. Furthermore, we propose a theoretical model based on the hydrodynamic theory that accurately predicts the heat flux across a wide range of operating parameters. Finally, we reveal size effects of the electric force on nucleation sites and rising bubbles, explaining the contrasting phenomena of bubble suppression and enhanced bubble detachment observed in electric field-enhanced boiling. The results of this study provide theoretical insight for optimizing phase-change heat transfer efficiency.

Pool boiling, recognized as one of the most efficient heat transfer mechanisms, has been widely employed in applications such as cooling of power electronics, heat exchangers, and thermal management systems in aerospace and aviation[1,2]. Over the past half-century, numerous techniques have been proposed to enhance pool boiling heat transfer efficiency, including the optimization of heated surface design[3] and the application of external fields[4]. Among these, electric field-enhanced pool boiling (referred to as EF pool boiling hereafter) has attracted considerable attention due to its remarkable capability to improve heat transfer performance[4–6]. The study of electric field-enhanced heat transfer dates back to the work of Chubb in 1916[7], which first demonstrated that applying an electric field can enhance heat transfer in fluids. However, systematic investigations of EF pool boiling emerged only in recent decades[4,8]. Most of the existing studies[9–12] reported an increased bubble nucleation density, reduced bubble detachment size, and enhanced bubble detachment frequency for EF pool boiling. The current consensus is that electric fields are particularly effective in enhancing heat transfer efficiency under high superheat conditions[13,14], with structured heated surfaces[11,12] or in microgravity environments[15–17].

In contrast to the extensive research on thermodynamic behaviour, relatively few studies have examined bubble dynamics and hydrodynamics in EF pool boiling[4,8]. Most of the existing research has focused on the hydrodynamic behaviour of single bubbles in electric fields[18,19]. Nevertheless, single-bubble dynamics often contradict those observed during multi-bubble pool boiling. For example, experiments[19,20] on single-bubble growth show that the application of electric fields suppressed bubble detachment, reduced bubble detachment frequency and prolonged bubble detachment time. It should be noted that this suppression of bubble detachment by electric fields predominantly occurs under quasi-static or slow bubble growth conditions. The mechanism underlying the contrasting phenomena of bubble suppression and enhanced bubble detachment observed in large-scale, multi-bubble EF pool boiling remains unclear[8].

So far, most EF pool boiling studies have focused on highly insulating liquids, while conductive fluids (usually electrical conductivity $\sigma > 10^{-5}\,\text{S}\cdot\text{m}^{-1}$) or leaky dielectric fluids (typically in the range of $\sigma \approx 10^{-5} \sim 10^{-12}\,\text{S}\cdot\text{m}^{-1}$)[21,22] have received little attention[6,23]. Some experiments[24,25] highlighted the potential of electric fields to mitigate heat transfer degradation in conductive media, suppress the Leidenfrost state in a variety of liquids, including conductive fluids such as deionized water and organic solvents (e.g., isopropanol, methanol, acetone). Although practical and safety challenges persist in applying electric fields to conductive liquids in traditional applications such as electronic cooling and microgravity

[1]National Microgravity Laboratory, Institute of Mechanics, Chinese Academy of Sciences, Beijing, China. [2]Institute for Multiscale Thermofluids, School of Engineering, The University of Edinburgh, Edinburgh, UK. [3]Department of Mechanical Engineering, University College London, Torrington Place, London, UK. [4]Key Laboratory of Thermo-Fluid Science and Engineering of Ministry of Education, School of Energy and Power Engineering, Xi'an Jiaotong University, Xi'an, Shaanxi, China. [5]School of Engineering Science, University of Chinese Academy of Sciences, Beijing, China. ✉e-mail: likai@imech.ac.cn; k.luo@ucl.ac.uk

environments, EF pool boiling shows unprecedented advantages in certain areas, including oil-water separation[26] and cathodic protection systems[27].

Thanks to advances in phase change multiphase flow models, numerical simulation has become an essential approach for investigating EF pool boiling. The foundations of electrohydrodynamic (EHD) multiphase flow simulation stem from the leaky dielectric model (LDM) proposed by Taylor[28], the electric force ($\mathbf{F}_e$) can be described as:

$$\mathbf{F}_e = \rho_e \mathbf{E} - \frac{1}{2}\mathbf{E}^2 \nabla\varepsilon + \nabla\left[\frac{1}{2}\mathbf{E}^2\rho\left(\frac{\partial\varepsilon}{\partial\rho}\right)\right], \tag{1}$$

where $\mathbf{E}$ is electric field strength, $\varepsilon$ is dielectric permittivity, $\rho_e$ and $\rho$ are electrical charge density and fluid density, respectively. The terms on the right-hand side represent the Coulomb force, polarization force, and electrostriction force, respectively. The Coulomb force dominates in conductive or leaky dielectric media, while the polarization force dominates in insulating liquids, the electrostriction force can usually be ignored in incompressible fluids. By coupling the LDM with the volume-of-fluid (VOF) method, some simulations are conducted for EF pool boiling in two-dimensions[29,30]. In recent years, the mesoscopic lattice Boltzmann method (LBM) has gained popularity for boiling simulations[31–33] due to its ability to directly model phase change using an equation of state, thus avoiding the need for empirical parameters[34]. Despite recent progress, knowledge gaps remain in understanding bubble dynamics and the effects of electric force in EF pool boiling, and large-scale three-dimensional simulations of EF pool boiling are still lacking.

In this work, we adopt a unified lattice Boltzmann framework (ULBM) with a central moment-based collision operator (CLBM)[35–38], integrating EHD multiphase flow model[39] to conduct large-scale three-dimensional simulations of EF pool boiling for leaky dielectric fluids. This study focuses on the effects of electric field strength, heating temperature, and electrical conductivity, intending to explore how electric fields influence bubble dynamics and enhance heat transfer efficiency. Based on Berghmans's model[40], we propose a theoretical model for the average heat flux in leaky dielectric fluids. Through systematic analysis of bubble evolution during pool boiling and detailed examinations of single-bubble boiling under electric fields, we reveal the size-dependent effects of electric force on generated bubbles and nucleation sites, providing a potential explanation for the paradoxical effects of electric fields on pool boiling bubbles.

## Results

A schematic representation of the simulation setup and its boundary conditions is shown in Supplementary Fig. 1. The simulation domain is a rectangular box with length ($L_x$) and width ($L_y$) of $0.08\,m$, height ($L_z$) of $0.04\,m$. The bottom surface is the heated surface, maintained at a constant temperature $T_h$. To initiate bubble nucleation, a small temperature disturbance with a standard deviation of $0.05T_h$ is imposed on the first grid layer above the heated surface. The top boundary is a no-slip wall, while the side boundaries are periodic. Initially, the lower two-thirds of the domain are filled with liquid, and an electric field is applied in the lower half of the domain ($H = L_z/2$). The electric field strength is defined as $E_0 = (\psi_1 - \psi_0)/H$, where $\psi$ denotes the electric potential. The saturation temperature is set to $T_s = 0.86T_c$, $T_c$ denotes the critical temperature of the fluid. The working fluids are leaky dielectric fluids whose density ratios approximate those of a water–vapor system. These properties are listed in Table 1, it should be noted that the electrical properties used in this study differ from those of realistic water–vapor systems. Higher liquid electrical permittivity and lower electrical conductivity were employed to highlight the effects of charge relaxation.

To quantify the effects of the electric field strength and superheating, we introduce several dimensionless numbers. The Jakob number, $Ja = c_{v,l}(T_h - T_s)/h_{fg}$, characterizes the ratio of sensible heat to latent heat. The electric capillary number, $Ca_e = \varepsilon_g L_z E_0^2/\gamma$, represents the ratio of the electric force to surface tension. Additionally, the Bond number, $Bo = \rho_l g L_z^2/\gamma$, quantifies the ratio of gravity to surface tension and is fixed

at 418 in all cases, corresponding to earth gravity ($g = 9.8\,m\cdot s^{-2}$). In the analyzes that follow, length and time are normalized by $l_r = \sqrt{\gamma/(g(\rho_l - \rho_g))}$ and $t_r = \sqrt{l_r/g}$, respectively. The effect of electrical conductivity is represented by the charge relaxation number, $T_\sigma = \varepsilon_l/(\sigma_l t_r)$, which indicates the timescale for free charges in a conducting or partially conducting fluid to redistribute and reach equilibrium.

Prior to starting, we note several assumptions. First, fluid properties and viscosities are assumed constant and independent of temperature. Second, the heated surface has a fixed contact angle of about 60°, and fluid-solid conjugate heat transfer is neglected. Third, the electrohydrodynamic multiphase flow is modeled using the leaky dielectric model, treating surface charges as volumetric charges within the interfacial diffusion layer. Previous studies[5,41] have shown that electric field enhancement of pool boiling mainly occurs under high superheat conditions. Thus, this study focuses primarily on transition boiling regime. It should be pointed out that, the charge relaxation time $t_e = \varepsilon_l/\sigma_l \sim 10^{-2} - 10^0$s is longer than both the charge diffusion time $t_{diff} = \mu_l l_r/\gamma \sim 10^{-4}$s and the charge convection time $t_{conv} = \mu_l/(\varepsilon_l E_0^2) \sim 10^{-2}$s, according to the non-dimensional analysis in Method section, the charge relaxation (related to $t_e/t_{diff} = 10^2 - 10^4$) and charge convection (related to electric Reynolds numbers $Re_e = t_e/t_{conv} = 10^0 - 10^2$) should be included. On the other hand, in a realistic water–vapor system, $t_e$ is several orders of magnitude lower than in our simulations, resulting in negligible charge convection and charge relaxation effects. To fully incorporate the leaky dielectric effects (charge convection and charge relaxation), higher liquid electrical permittivity and lower electrical conductivity were employed in our simulations. Details of the numerical models and their validations are described in the accompanying Method section. The chosen mesh resolution is $dx = L_z/400 = 100\mu m$, resulting in over 250 million computational cells. It should be noted that the condition $L_{x,y,z} \gg l_r$ is well satisfied with current mesh resolution, indicating that the computational domain dimensions are sufficiently large compared to the bubble length scale. Consequently, our simulations can resolve a substantial number of bubbles (up to several hundred). The code is parallelized using MPI, and a typical case (modeling the dynamic behaviour of EF pool boiling for $T^* = T/t_r \approx 60$) requires about 35,000 CPU running over 20 h.

### Effects of electric field strength

We begin by simulating pool boiling with a fixed superheated wall temperature $T_h = 1.3T_s$, corresponding to $Ja = 0.3$. The ratio of electrical conductivity between liquid and gas phases ($\sigma_l/\sigma_g$) is set to $2.5 \times 10^{10}$, giving $T_\sigma = 6.4$. Figure 1a, b compare experimental snapshots[41] and our simulation results for conventional and EF pool boiling, respectively. It is evident that applying an electric field substantially increases the number of generated bubbles and reduces their size. Moreover, there is a significant increase in the number of nucleation sites above the heated surface, accompanied by a decrease in their radius, this effect that has rarely been directly observed experimentally due to its highly transient nature. We analyzed the normalized bubble size distribution ($R_{<>}^*$) for all bubbles produced during the simulation. As shown in Fig. 1c, the size and number of generated bubbles follow a normal distribution, consistent with Ünal's classic theory[42]. For the pool boiling case with the strongest electric field strength ($Ca_e = 1.6$), the mean bubble radius decreases by 50% compared to conventional pool boiling, and the number of bubbles increases by 5 times. Assuming that the mean bubble radius is on the same order as the most dangerous wavelength of the vapor-liquid interface, by extending Berghmans's hydrodynamic model[40], the average bubble radius can be estimated using Helmholtz instability theory modified to incorporate electric field effects:

$$\bar{R}_{theory}^* = a\lambda_d^* = \frac{a6\pi 2^{0.5}}{Ca_e f(T_\sigma) + \sqrt{3Bo + (Ca_e f(T_\sigma))^2}}, \tag{2}$$

compared to original Berghmans's model[40], we introduce an adjustment function $f(T_\sigma) = 1 + 100/T_\sigma + 4\mathrm{Ln}(1/T_\sigma + 1)$ to account for the influence of electrical conductivity. It should be pointed out that the proposed adjustment function is only validated for leaky dielectric and insulating fluids. For purely insulating fluids (with infinite $T_\sigma$), Eq. (2) reduces to the original hydrodynamic model[40]. This model also can be reduced to the classic hydrodynamic relation $\bar{R}^*_{theory} \sim \mathrm{Bo}^{-0.5}$ of Zuber[43], in the limit $\mathrm{Ca}_e = 0$. The fitting parameter $a = 0.98$ is selected based on Berghmans's[40] original model constant $a = 1.0$, with a slight adjustment to ensure accurate prediction of bubble sizes under electrohydrodynamic conditions. The predicted $\bar{R}^*_{theory}$ are shown as vertical dashed lines in Fig. 1c and agree well with our numerical results for both conventional and EF pool boiling.

Figure 1(d) plots the temporal evolution of both the number of bubbles and their average radius ($\bar{R}^*_{<>}$) in Fig. 1d during the simulation. Our results indicate that the bubble growth period (defined as the time from initial heating to bubble generation) shortens with increasing $\mathrm{Ca}_e$, consistent with findings by Kweon[10] and Diao et al.[44]. Notably, a higher number of nucleation bubbles reduces the thermal resistance between the heated wall and the boiling liquid, implying enhanced heat transfer efficiency. We define

the transient heat flux as:

$$Q_h = \frac{1}{L_x L_y} \iint \left[ -\lambda \left( \frac{\partial T}{\partial z} \right) \Big|_{(z=H_w)} \right] dx dy, \tag{3}$$

where $H_w$ denotes the location of heat wall surface. The evolution of non-dimensionalised transient heat flux $\tilde{q} = Q_h/q_r$ is plotted in Fig. 1e, the term $q_r = h_{fg}\rho_g\sqrt{g\gamma(\rho_l - \rho_g)}$ stands for the Zuber's correlation[43]. As shown in figure, after heating, due to the evaporation of liquid above heated surface, $\tilde{q}$ notably decreases within a short period of time ($T^* < 10$, gary background regime), then $\tilde{q}$ gradually increases as nucleation begins and approaches a quasi-steady state (e.g., after $T^* > 35$), exhibiting the typical heating process for transition boiling. The increasing period of $\tilde{q}$ implies the nucleation stage of boiling bubbles, and $\tilde{q}$ reaches to its maximum value when the generated bubbles detach from the heated plate. For higher $\mathrm{Ca}_e$, $\tilde{q}$ reaches its peak value more rapidly, as pointed by the black dots in Fig. 1e, which implies a faster bubble detachment frequency. Considering $\mathrm{Ja} = 0.3$ close to the CHF temperature (as proved in the next section), similar to the derivation of Zuber[43] and Berghmans[40], the average heat flux $\bar{q}$ can be achieved by hydrodynamic theory, implementing the wavelength as introduced in Eq. (2), we have:

$$\bar{q} = \frac{\pi\sqrt{\rho_g}h_{fg}}{10q_r} \left( \frac{\gamma\rho_g g}{3} \right)^{0.25} \left[ \frac{\mathrm{Ca}_e f(T_\sigma)}{\sqrt{3\mathrm{Bo}}} + \left[ \frac{(\mathrm{Ca}_e f(T_\sigma))^2}{3\mathrm{Bo}} + 1 \right]^{0.5} \right]^{0.5}. \tag{4}$$

Equation (4) is plotted in the inset figure of Fig. 1e. It can be found that the proposed theoretical equation agreed well with $\bar{q}$ for a wide range of $\mathrm{Ca}_e$. It should be noted that the above hydrodynamic theoretical model is derived based on flat plate assumption and has not been validated for complex geometrical heating surfaces. In the following, Eq. (4) will be further validated extensively by simulation results from varying $T_\sigma$ and single bubble boiling cases.

**Table 1 | Fluid properties used in the current study**

| | Liquid phase (l) | Vapour phase (g) |
|---|---|---|
| Density $\rho$ (kg · m$^{-3}$) | 739.7 | 36.5 |
| Viscosity $\mu$ (Pa · s) | 0.01 | 0.003 |
| Specific heat capacity $c_v$ (J · kg$^{-1}$ · K$^{-1}$) | 3070.83 | 2808.73 |
| Thermal conductivity $\lambda$ (W · m$^{-1}$ · K$^{-1}$) | 8.5 | 0.163 |
| Permittivity $\varepsilon$ (F · m$^{-1}$) | $7.08 \times 10^{-10}$ | $8.85 \times 10^{-12}$ |
| Electrical conductivity $\sigma$ (S · m$^{-1}$) | $7.5 \times 10^{-9} \sim 3 \times 10^{-10}$ | $3 \times 10^{-19}$ |
| Specific enthalpy $h_{fg}$ (kJ · kg$^{-1}$) | 1257.9 | |
| Surface tension $\gamma$ (N · m$^{-1}$) | 0.025 | |

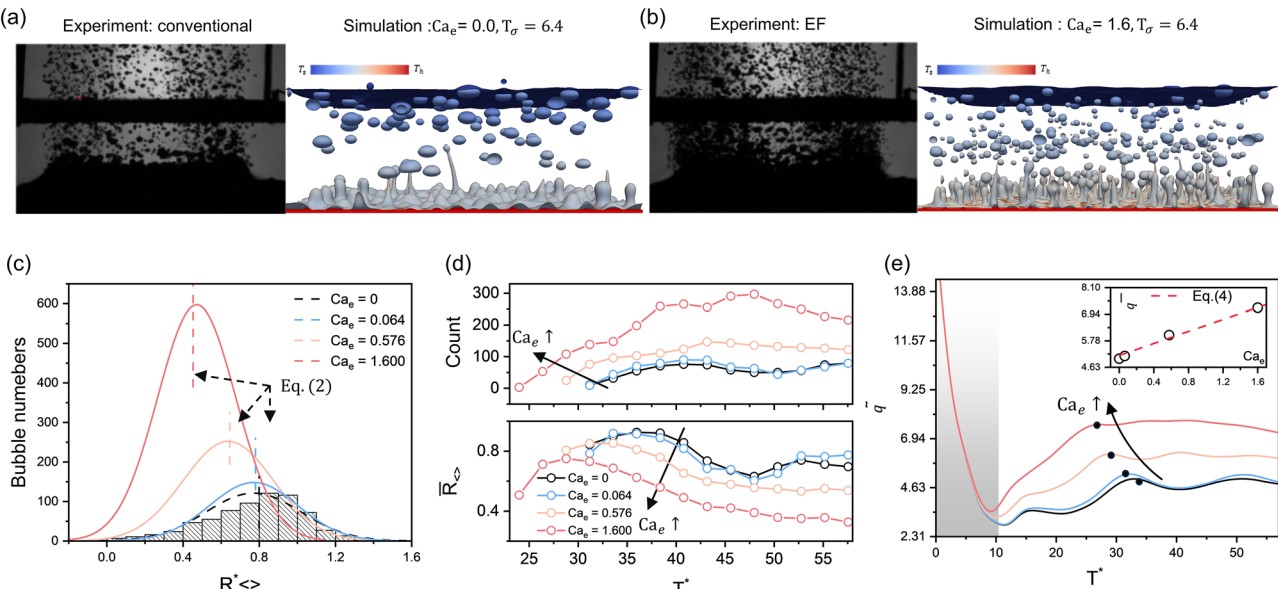

**Fig. 1 | Effects of electric field strength on electric field-enhanced pool boiling.** Selected snapshots comparing experimental results[41] (left gray plot) with simulation results (right color plot, in temperature-mapped) for (**a**) conventional pool boiling and (**b**) EF pool boiling. **c** Normalized bubble size distribution for the conventional pool boiling (column data with fitted dashed lines) and EF pool boiling with varying electric capillary number $\mathrm{Ca}_e$ (solid fitted lines). Comparison of the time-resolved evolution of (**d**) total bubble count (top figure) and averaged bubble radius (bottom figure), (**e**) transient heat flux $\tilde{q}$ for varying $\mathrm{Ca}_e$. The inset in (**e**) shows averaged heat flux $\bar{q}$ as a function of $\mathrm{Ca}_e$.

## Effects of superheated temperature

Next, we examine how the electric field enhances pool boiling at different superheated temperatures. Figure 2a shows the dynamic evolution of pool boiling at $Ca_e = 1.6$ for various Ja. Qualitatively, increasing Ja significantly prolongs the bubble growth period and increases bubble size. Notably, for conventional pool boiling at Ja = 0.35, the boiling regime shifts from transition to film boiling (see Supplementary Fig. 5). In contrast, EF pool boiling at higher Ja (e.g., Ja = 0.35, 0.4 and 0.5) exhibits a reverse transition, namely from film boiling back to transition boiling. For EF pool boiling at Ja = 0.5, we initially observe film boiling behaviour ($T^* < 30$). As the liquid film above the heated surface becomes disturbed by the electric force, the system reverts to transition boiling characteristics, gradually forming nucleation sites ($T^* = 38.4$) and generating bubbles ($T^* = 48$). Figure 2b shows the evolution of bubble counts with time. For conventional pool boiling, the bubble growth period varies non-monotonically with Ja, initially decreasing and then increasing, with the shortest period occurring at the CHF condition (Ja = 0.3). For the cases of EF pool boiling, the bubble growth period increases monotonically with Ja, consistent with the qualitative observations in Fig. 2a.

As shown in Fig. 2c, for the EF pool boiling case at Ja = 0.5, the transient heat flux $\tilde{q}$ initially exhibits values similar to those observed during film boiling, then rapidly increases as bubble nucleation commences. The inset of Fig. 2c indicates that the critical heat flux (CHF) is reached at Ja = 0.3 for conventional pool boiling, after which $\bar{q}$ decreases rapidly with increasing Ja. In the transition boiling regime (Ja>0.3), $\bar{q}$ can be predicted by the hydrodynamic model (Eq. (4)) with a correction term[45]:

$$\bar{q}(\text{Ja}) = f(\text{Ja})\bar{q} = \left(1 - m\left(\frac{\text{Ja} - \text{Ja}_{chf}}{\text{Ja}_{chf}}\right)^n\right)\bar{q}, \quad (5)$$

where $\text{Ja}_{chf} = 0.3$ is the critical Jakob number, $m$, $n$ are fitting parameters. For conventional pool boiling, the best fit parameters are $m = 50$ and $n = 2.6$, for EF pool boiling $m = 0.657$ and $n = 1.5$. The corrected equation and our simulation results are plotted in the inset of Fig. 2c. As illustrated in

the figures, the fitting parameters result in high prediction accuracy within the investigated parameters. It shows that for EF pool boiling at Ja = 0.3, $\bar{q}$ is 44.5% higher than the CHF of conventional pool boiling. Moreover, when Ja>0.3, $\bar{q}$ for EF pool boiling is significantly improved, for example, it is 168% higher than in conventional pool boiling at Ja = 0.35. Figure 2d compares the size distribution of generated bubbles. For high superheat cases (Ja> 0.3), the number of generated bubbles in EF pool boiling is an order of magnitude greater than in conventional pool boiling. Additionally, the mean bubble radius decreases as Ja increases for both conventional and EF pool boiling, consistent with the qualitative observations.

## Effects of electrical conductivity

We then focus on the effect of electrical conductivity on EF pool boiling. Due to the computational complexity involved in simultaneously solving the charge convection-diffusion, Poisson, and Navier-Stokes equations, previous numerical studies on EF pool boiling have often neglected the Coulomb force. Moreover, insulating fluids are commonly used as working fluids in pool boiling experiments[4,5]. To address this gap and investigate EF pool boiling in leaky dielectric fluids, we examined four different conductivity ratios at $Ca_e = 1.6$. By varying the liquid conductivity $\sigma_l$, we increased $\sigma_l/\sigma_g$ from $10^9$ to $2.5 \times 10^{10}$, corresponds to $T_\sigma$ decreases from 160 to 6.4.

Figure 3a illustrates the qualitative evolution of the boiling process at constant $Ca_e$. It is evident that a lower $T_\sigma$ (i.e., higher liquid electrical conductivity) leads to denser bubbles and smaller nucleation sites. For the lowest $T_\sigma = 6.4$, quantitative results indicate that the number of bubbles is 5–6 times greater than for the highest $T_\sigma = 160$ (see Fig. 3b). The theoretical average bubble radius $\bar{R}^*_{theory}$ for various $T_\sigma$ are also plotted in Fig. 3b, showing an approximate 50% increase in average bubble radius as $T_\sigma$ increases from 6.4 to 160. Figure 3c presents the transient evolution of the heat flux $\tilde{q}$, it can be observed that the maximum $\tilde{q}$ increases with $\sigma_l$, and for EF pool boiling at $T_\sigma = 6.4$, $\bar{q}$ is 28.5% higher than $T_\sigma = 160$. However, increasing conductivity has little effect on bubble detachment frequency, as the time at which $\tilde{q}$ reaches its maximum value remains nearly unchanged

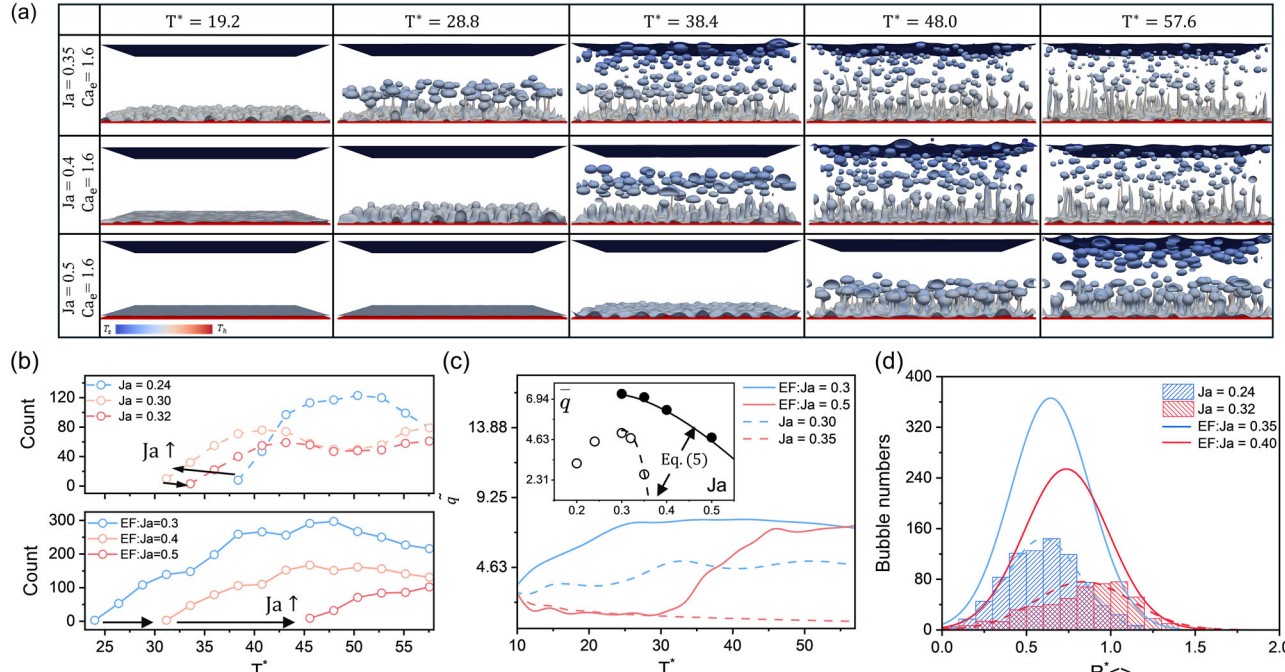

**Fig. 2 | Effects of superheat temperature on electric field-enhanced pool boiling. a** Snapshots of EF pool boiling process at electric capillary number $Ca_e = 1.6$ with different Jakob number Ja. **b** Transient evolution of the total bubble count for conventional pool boiling (top figure) and EF pool boiling (bottom figure). **c** Transient evolution of heat flux $\tilde{q}$ for conventional pool boiling (dashed lines) and

electric field-enhanced pool boiling (solid lines), with the inset showing the value of time averaged heat flux $\bar{q}$, compared with the theory predictions from Eq. (5). **d** Averaged bubble radius distributions for conventional pool boiling (column data with fitted dashed lines) and EF pool boiling (solid lines).

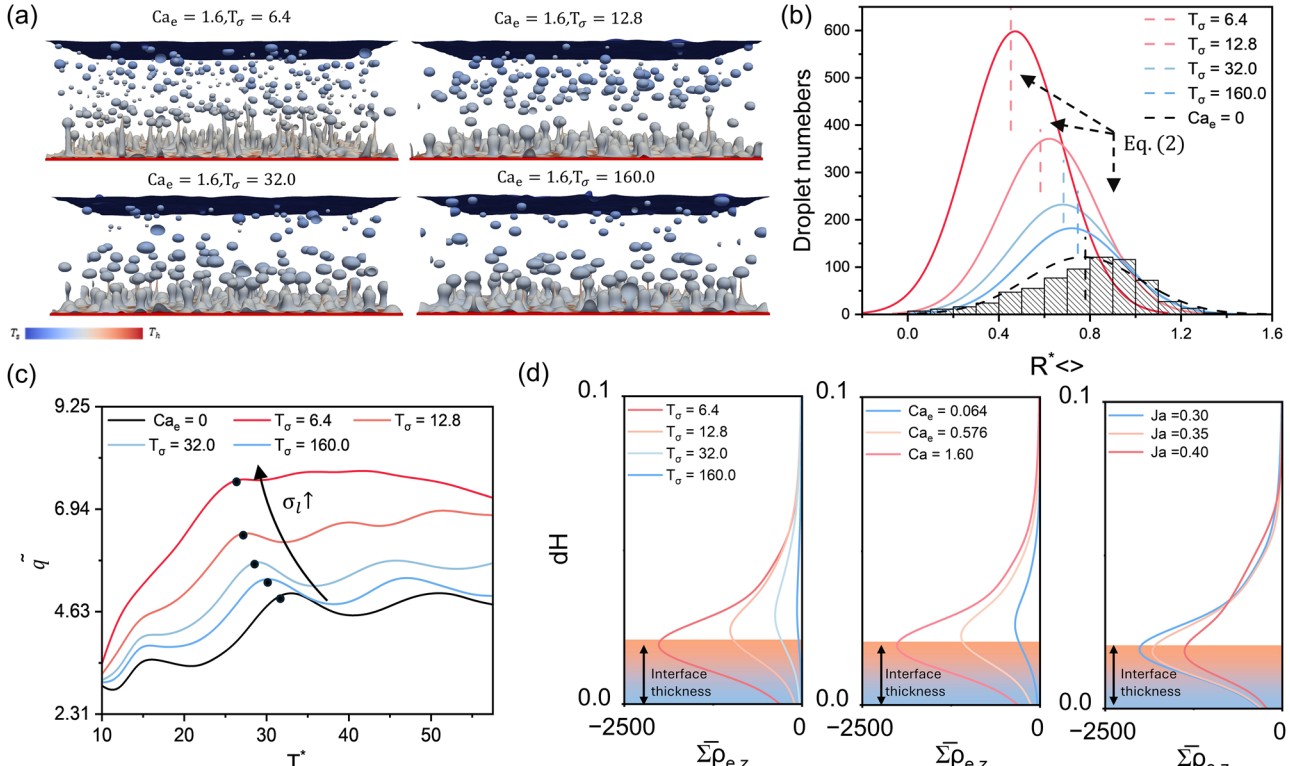

**Fig. 3 | Effects of electrical conductivity on electric field-enhanced pool boiling and charge density distribution under various operating conditions. a** Selected snapshots of EF pool boiling at varying charge relaxation number $T_\sigma$, along with the corresponding quantitative comparison: (**b**) bubble radius distributions, with the vertical dashed lines represent the theoretical prediction for the bubble radius (Eq.

(2)) and (**c**) transient evolution of heat flux $\tilde{q}$. Subfigures in (**d**) present accumulated charge density $\sum \bar{\rho}_{e,z}$ near the bottom surface at different $T_\sigma$, electric capillary number $Ca_e$ and Jakob number Ja, respectively, the colored region in the figure represents the phase interface.

across all cases (see black dots in Fig. 3c). Similar observations arise from the qualitative evolution of the single bubble boiling process (see Supplementary Fig. 6).

To further elucidate the mechanisms underlying electric field-enhanced pool boiling in leaky dielectric fluids, we examined the accumulated charge density $\sum \bar{\rho}_{e,z}$ close to the superheated surface (where dH = $(Z - H_w)/L_z$), where the charge accumulation effects are strongest. As shown in Fig. 3d. The accumulated charge density is defined as:

$$\sum \bar{\rho}_{e,z} = \frac{1}{L_x L_y} \iint \rho_{e,z} dx dy. \qquad (6)$$

Compared to the lowest $T_\sigma$ case, the free charge is negligible at the highest $T_\sigma$. In other words, the boiling liquid at the highest $T_\sigma$ behaves like a perfect dielectric fluid, and the electric field's influence stems primarily from the polarization force. Under these conditions, the average heat flux increases by about 15% compared to conventional pool boiling, aligning well with experimental results for EF pool boiling in insulating media[4,12,13]. In accordance with Gauss's law, $\sum \bar{\rho}_{e,z}$ increases with increasing $Ca_e$. Moreover, $\sum \bar{\rho}_{e,z}$ slightly decreases as Ja increases, primarily because higher temperatures lower the gas-liquid density ratio. Since electrical conductivity and dielectric permittivity are directly related to fluid density, a reduced density ratio leads to smaller gradients in conductivity and permittivity, thus lowering the charge density $\rho_e$ at the interface. It is also worth noting that we employ a diffuse-interface-based multiphase flow model. As the liquid adjacent to the superheated surface is not fully saturated during the boiling process, $|\sum \bar{\rho}_{e,z}|$ initially increases with height, reaches a maximum near the vapor-liquid interface, and then gradually decreases.

## Discussion

Based on the results presented above, it can be concluded that increasing $Ca_e$ or decreasing $T_\sigma$ (i.e., increasing $\sigma_1$) significantly enhances heat transfer efficiency. This is particularly important for pool boiling under high Ja conditions, where the application of an electric field can mitigate the heat transfer deterioration associated with film boiling. Moreover, compared to large electric field strength, increasing $\sigma_1/\sigma_g$ is a more cost-effective method for improving heat transfer, as it can be achieved by adding ions to the liquid phase[46].

In the following, we focus on analyzing quantitative data that are challenging to measure experimentally, such as wetting areas and the force distribution acting on generated bubbles. Figure 4a shows the growing nucleation sites on the heated surface for EF pool boiling at Ja = 0.3. The liquid-gas interface is colored by the magnitude of the electric force, and the dark gray background represents the solid-liquid interface. A pronounced size-dependent effect of the electric force distribution is evident. For continuous nucleation sites (indicated by blue arrows and insert figure in blue bracket), the electric force concentrates along the triple contact line (gas-liquid-solid) near the base of the site, consistent with our earlier analysis suggesting that the electric field exerts a compressive effect on growing sites. For smaller, dispersed nucleation sites (indicated by pink arrows and insert figure in pink bracket), significant electrical effects are observed across the entire site, inhibiting small bubble growth. As a result, at higher $Ca_e$, these smaller dispersed sites are almost completely absent. Additionally, noticeable electric force appears in regions where nucleation sites are shrinking (green arrows and insert figure in green bracket), indicating that the electric field promotes the fragmentation of these sites and the formation of monodisperse bubbles.

Besides, it can be observed that higher $Ca_e$ and $\sigma_1/\sigma_g$ result in a stronger electric force magnitude and larger wetted area ($S_{wet}$, i.e., the solid-

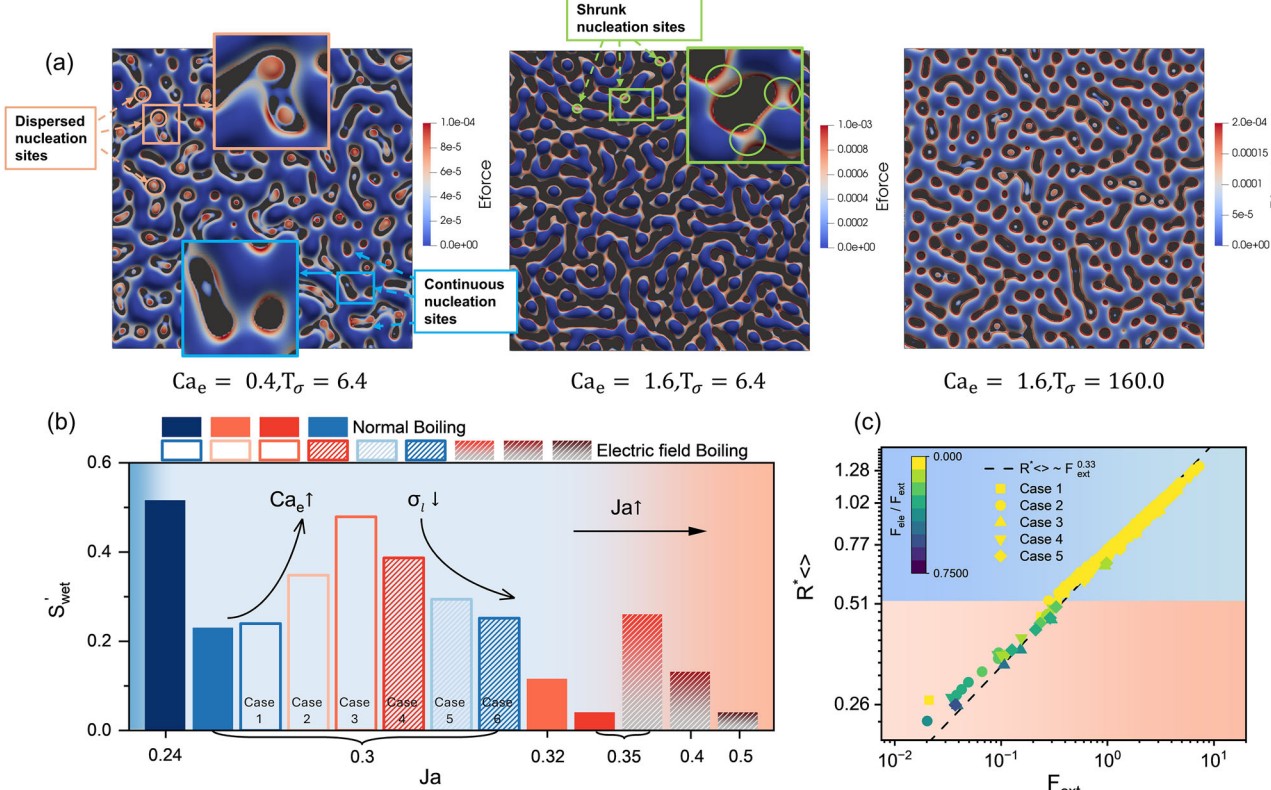

**Fig. 4 | Analysis of electric force distribution, average wetting area, and total forces acting on bubbles in electric field-enhanced pool boiling. a** Selected snapshots showing the distribution of electric force on the gas-liquid interface when $T^* = 19.2$, the black areas in the images represent the solid-liquid interfaces, sub-figures within the bracket show a zoomed-in view of the electric force distribution. **b** Normalized wetted surface area for conventional pool boiling (filled bars in a chart) and EF pool boiling (hollow and dashed bars in the chart). **c** External forces magnitude $F_{ext}$ as a function of normalized bubble size $R^*_{<>}$ at the instant after the first bubble shedding, where the dashed line represents the power law fit equation. The symbols in (**c**) are color-mapped by the electric force ratio $F_{ele}/F_{ext}$, the upper blue region represents the buoyancy dominated regime, while the red region corresponds to the electric force dominated regime.

liquid contact area). We computed the normalized wetted surface area ($S'_{wet} = S_{wet}/(L_x L_y)$) over time for all cases, as shown in Fig. 4b. The results indicate that $S'_{wet}$ decreases with increasing Ja for both conventional and EF pool boiling. At the same Ja, $S'_{wet}$ increases with higher $Ca_e$ and $\sigma_l$, consistent with our qualitative observations. For cases with Ja = 0.35, $S'_{wet}$ for EF pool boiling is an order of magnitude larger than that of conventional pool boiling, reflecting the transition from film boiling to transition boiling under the applied electric field. Recently, Graffiedi et al.[47] employed high-resolution optical diagnostics to investigate EF pool boiling of the dielectric fluid. Their study similarly observed reductions in continuous bubble footprints and increases in bubble departure frequency with applied electric fields. These experimental results align closely with our simulation findings, providing strong support for the above analysis regarding bubble nucleation sites.

As illustrated in Fig. 4a, the electric force acting on growing bubbles exhibits a significant size dependence. To further investigate this effect, we recorded the external forces magnitude ($F_{ext}$) acting on dispersed rising bubbles for various combinations of $Ca_e$ and $\sigma_l/\sigma_g$ (cases 1 ~ 5 in Fig. 4b). As shown in Fig. 4c, the forcing effects on the bubbles can be categorized into two regions based on normalized bubble size $R^*_{<>} = R_{<>}/l_c$: an electric-force-dominated region ($R^*_{<>} < 0.51$) and a buoyancy-dominated region ($R^*_{<>} > 0.51$). Since the external force on the bubbles is the sum of the electric force and buoyancy, i.e., $F_{ext} = F_{ele} + F_b$, when electric force ratio $F_{ele}/F_{ext}$ approaches zero, the external force is governed by buoyancy. According to the relation $F_{ext} \sim F_g = 4/3\pi R^3_{<>}\Delta\rho g$, $R^*_{<>}$ follows a power law of external force with an exponent of 1/3 (indicated by the dashed line in Fig. 4(c)). For smaller bubbles, however, the contribution of $F_{ele}$ becomes significant, causing deviations from power-law relationship.

To clarify the mechanisms of enhanced bubble departure observed in pool boiling, single-bubble simulations are introduced to elucidate fundamental bubble dynamics under electric fields. The simulation parameters match those in Table 1, except that a small bubble is initially introduced at the surface to serve as a nucleation site. We tested cases with different $Ca_e$ and $T_\sigma$. The qualitative results of the bubble growth process are shown in Fig. 5a, where the arrows represent the electric force (comprising both Coulomb and polarization contributions). At the early stage of bubble growth ($T^* = 7.2$), a continuous gas film is observed around the main bubble, indicating transition boiling conditions. The electric force concentrates near the triple contact line (labeled "$N$" in the figure) and points toward the bubble interior. This compression accelerates neck constriction and promotes bubble growth. Simultaneously, the compressive force component acting on the continuous liquid film leads to neck constriction, resulting in the formation of dispersed bubbles. Moreover, the substantial electric force along the triple contact line prevents the gas-liquid interface from expanding, thereby suppressing film boiling.

Examining the effect of the electric force in more detail, its vertical component inhibits bubble growth, whereas its horizontal component exerts a strong compressive effect, elongating the bubbles. According to Zuber's model[43], bubble growth during pool boiling is driven by Helmholtz instabilities, the shear velocity between the vapour jet and ambient liquid is:

$$\Delta u = \left[\frac{\rho_l - \rho_g}{\rho_l \rho_g}\right]^{0.5} \left[\frac{2\gamma}{D_j}\right]^{0.5}, \qquad (7)$$

where $D_j$ is the vapor jet diameter. Elongated bubbles have smaller vapor jet diameters and thus higher shear velocities, facilitating bubble breakup. This

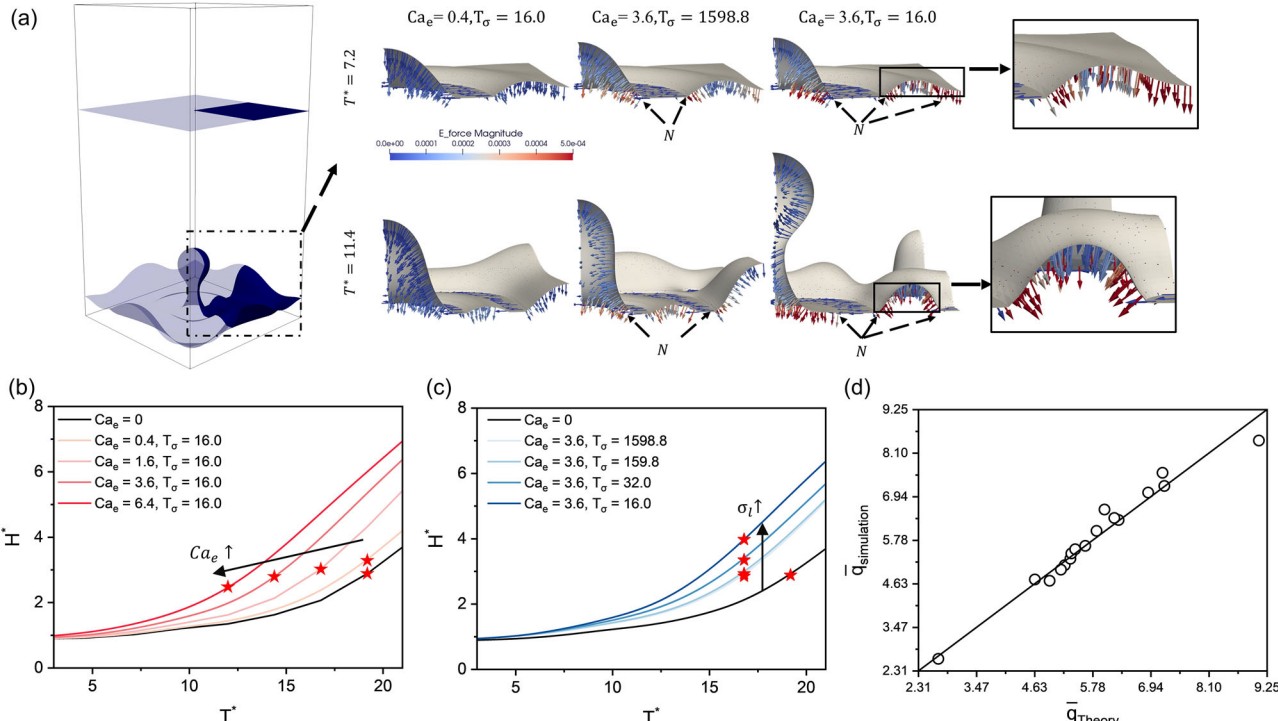

**Fig. 5 | Analysis of single-bubble nucleation pool boiling. a** Schematic of the simulation domain for single bubble nucleation pool boiling and the time evolution of bubbles growing. Arrows in the figure indicate the direction and magnitude of the electric force and subfigures within the bracket show a zoomed-in view of the forcing distribution on continuous liquid film. Evolutions of the bubble front position for cases with different (**b**) electric capillary number $Ca_e$ and (**c**) charge relaxation number $T_\sigma$, the pentagrams in the figures represents the instant of bubble detachment. **d** A comparison between theoretical prediction and simulation results of averaged heat flux $\bar{q}$.

result addresses a longstanding debate regarding whether electric fields increase or decrease bubble detachment time. For freely growing bubbles[9,10], the electric field enhances detachment due to its compressive effect. In contrast, for stationary bubbles[19,20,48], the electric field suppresses bubble departure because of its vertical force component. Additionally, we tracked the evolution of the bubble front under different $Ca_e$ and $T_\sigma$, as shown in Fig. 5 b, c, respectively. The pentagrams indicate the moments of bubble detachment. The bubble growth velocity increases with $Ca_e$ and decreases with $T_\sigma$, aligning with previous predictions. Notably, as $Ca_e$ increases, the bubble growth period shortens and detachment frequency rises, whereas changes in $T_\sigma$ have little effect on detachment time. On the other hand, higher $Ca_e$ or lower $T_\sigma$ promotes the formation of more dispersed bubbles (see Supplementary Fig. 5). Finally, we compared the time-averaged heat flux $\bar{q}$ for all simulation cases with the theoretical models (Eq. (4) and Eq. (5)) in Fig. 5d. Our proposed theoretical model accurately predicts $\bar{q}$ over a wide range of operating parameters for both pool boiling and single-bubble cases.

In summary, we developed a novel mesoscopic phase-change electrohydrodynamic (EHD) lattice Boltzmann method (LBM) model and conducted a parametric investigation of electric field-enhanced (EF) pool boiling in leaky dielectric fluids. Our results indicate that increasing both electric field strength and liquid electrical conductivity can significantly enhance boiling heat transfer. Notably, the application of an electric field effectively mitigates heat transfer deterioration caused by film boiling, driving transitions from film boiling to transition boiling at Ja=0.5. It's observed that increasing $Ca_e$ or decreasing $T_\sigma$ reduces the bubble detachment radius and increases the wetted surface area on the heated wall, thereby improving heat transfer efficiency. Building on hydrodynamic theory, we extended Berghmans's[40] model for heat flux to cover a wide range of $T_\sigma$ and Ja, and the extended theoretical model agrees well with our simulation results. Furthermore, we revealed the size-dependent effects of the electric force on both nucleation sites and rising bubbles. For large nucleation sites,

the electric force is concentrated at the contact lines, exerting a compressive effect that prevents vapor film detachment. For smaller nucleation sites, the electric force acts on the entire site, inhibiting bubble growth. As for rising bubbles, the electric force primarily drives small bubbles ($R^*_{<>}<0.51$), while buoyancy dominates for larger bubbles ($R^*_{<>}>0.51$). It is worth noting that, owing to the substantial computational complexity and the extensive numerical grid requirements (over 250 million cells), a comprehensive exploration was limited in the present study. Consequently, future experimental and numerical investigations focusing on realistic leaky dielectric fluids across a wider range of operating parameters are essential. Such studies would further enhance our understanding of electric field effects on pool boiling phenomena and validate the applicability and robustness of our proposed theoretical model for heat flux.

## Methods
### Phase change multiphase flow
To accurately simulate the EF pool boiling for leaky dielectric fluids, the coupling among multiphase fluid dynamics, phase-change heat transfer, and electric field interactions should be considered. The governing equations for the fluid flow are described as NS equations:

$$\frac{\partial \rho}{\partial t} + \nabla \cdot (\rho \boldsymbol{u}) = 0,$$

$$\frac{\partial (\rho \boldsymbol{u})}{\partial t} + \nabla \cdot (\rho \boldsymbol{uu}) = -\nabla P + \nabla \cdot \left(\mu \left(\nabla \boldsymbol{u} + \nabla \boldsymbol{u}^T\right) + \left(\mu_b - \frac{2}{3}\mu\right)(\nabla \cdot \boldsymbol{u})\boldsymbol{I}\right) + \mathbf{F},$$

$$\text{(8)}$$

where $\mathbf{F} = \mathbf{F_e} + \mathbf{F_{int}} + \mathbf{F_b}$, the mesoscopic pseudopotential force $\mathbf{F_{int}}$ stands for the interaction between the liquid and gas phases. $\mathbf{F_b} = -(\rho - \rho_{avg})g\boldsymbol{j}$ is the buoyancy, and $\rho_{avg}$ is the average density of the liquid and vapor phases, $\mu_b$ is fluid bulk viscosity.

To solve above NS equations, we constructed a cascaded lattice Boltzmann collision operator (CLB) within the ULBM framework[35] using a

non-orthogonal moment set proposed by Fei et al.[37,38]. The D3Q19 discrete velocity model is adopted to capture the hydrodynamic behaviours in three-dimensions. The general evolution equation for ULBM(CLB) model can be expressed as:

$$
f_i(\boldsymbol{x} + \boldsymbol{e}_i \Delta t, t + \Delta t) \equiv f_i^*(\boldsymbol{x}, t) = \mathbf{M}^{-1}\mathbf{N}^{-1}(\mathbf{I} - \mathbf{S})|\tilde{\mathrm{T}}_i\rangle
$$
$$
+ \mathbf{M}^{-1}\mathbf{N}^{-1}\mathbf{S}|\tilde{\mathrm{T}}^{\mathrm{eq}}\rangle + \mathbf{M}^{-1}\mathbf{N}^{-1}(\mathbf{I} - \mathbf{S}/2)|C_i\rangle, \quad (9)
$$

where $f_i$ and $f_i^*$ indicate pre-collision and post-collision discrete distribution functions, respectively. $i = 0 \ldots 19$ and $|\cdot\rangle$ denotes a 19-column vector. $|\tilde{\mathrm{T}}_i\rangle$, $|\tilde{\mathrm{T}}^{eq}\rangle$ and $|C_i\rangle$ stand for the discrete distribution moment set, discrete equilibrium moment set and discrete forcing moment set in the central moment space, respectively. I is the unit matrix and $\mathbf{S}$ is the relaxation matrix. In above evolution equation for ULBM(CLB), the transformation matrix $\mathbf{M}$ is adopted to transform the distribution functions ($f_i$) to their raw moments ($\mathrm{T}_i$). The shift matrix $\mathbf{N}$ is used to shift the raw moments ($\mathrm{T}_i$) into the central moments ($\tilde{\mathrm{T}}_i$), and the transformation/shift can be expressed as:

$$
f_i = \mathbf{M}^{-1}\mathrm{T}_i = \mathbf{M}^{-1}\mathbf{N}^{-1}\tilde{\mathrm{T}}_i. \quad (10)
$$

The corresponding shift matrix $\mathbf{N}$, the non-orthogonal transformation matrix $\mathbf{M}$ and the explicit expression for $|\tilde{\mathrm{T}}_{eq}\rangle$, $|C_i\rangle$ can be found in our previous studies[35,36]. With the Chapman-Enskog (CE) analysis, the above ULBM (CLB) model can reproduce the macroscopic N-S equations in Eq. (8).

To simulate the multiphase flow, the extended combined pseudopotential (ECP) model proposed by Wang et al.[36] is used, where the pseudopotential force $\mathbf{F_{int}}$ can be written as:

$$
\mathbf{F_{int}} = -\left(\frac{\xi}{2} - \frac{k}{6}\right)\sum_i w\left(|\boldsymbol{e}_i|^2\right)\psi^2(\boldsymbol{x}+\boldsymbol{e}_i)\boldsymbol{e}_i
$$
$$
- \left(1 - \xi + \frac{k}{3}\right)G\psi(\boldsymbol{x})\sum_i w\left(|\boldsymbol{e}_i|^2\right)\psi(\boldsymbol{x}+\boldsymbol{e}_i)\boldsymbol{e}_i - k\frac{Gc^4}{6}\nabla^2\psi\nabla\psi, \quad (11)
$$

where $k$ and $\xi$ are free parameters to adjust surface tension and thermodynamic consistency, respectively. $G = -1$ is the interaction strength, $\boldsymbol{e}_i$ is the discrete velocity and $w(|\boldsymbol{e}_i|^2)$ are weights for the D3Q19 model, $\psi$ is the square-root pseudopotential:

$$
\psi = \sqrt{\frac{2(\mathrm{P_{EOS}} - \rho c_s^2)}{Gc^2}}, \quad (12)
$$

where $c = 1$ is the lattice constant, $c_s^2 = 1/3$ is the lattice sound speed. $\mathrm{P_{EOS}}$ is the pressure calculated by the equation of state (EOS). To simulate the multiphase flow with phase change phenomena, we use the Peng–Robinson EOS, where:

$$
\mathrm{P_{EOS}} = \frac{\rho RT}{1 - b\rho} - \frac{a\varphi(T)\rho^2}{1 + 2b\rho - b^2\rho^2}, \quad (13)
$$

where $a = 0.4572R^2\mathrm{T}_c^2/\mathrm{P}_c$, $b = 0.0778R\mathrm{T}_c/\mathrm{P}_c$, and $\varphi(\mathrm{T}) = \left[1 + \left(0.37464 + 1.54226\omega - 0.26992\omega^2\right)\left(1 - \sqrt{\mathrm{T}/\mathrm{T}_c}\right)\right]^2$, $\mathrm{P}_c$ and $\mathrm{T}_c$ stand for the critical pressure and critical temperature, respectively. In this work, we set $R = 1$, $\omega = 0.344$, $a = 2/49$ and $b = 2/21$, with the corresponding $\mathrm{T}_c = 0.0729$. After streaming of LB evolution, the macroscopic variables can be expressed as:

$$
\rho = \sum_i f_i, \rho\boldsymbol{u} = \sum_i f_i\boldsymbol{e}_i + \frac{\Delta t\mathbf{F}}{2}. \quad (14)
$$

Inspired by Li et al.[49], the temperature field for the liquid-vapor phase-change can be written as:

$$
\frac{\partial \mathrm{T}}{\partial t} = -\boldsymbol{u}\cdot\nabla\mathrm{T} + \frac{1}{\rho c_v}\left(\lambda\nabla^2\mathrm{T} + \nabla\lambda\cdot\nabla\mathrm{T}\right) - \frac{\mathrm{T}}{\rho c_v}\left(\frac{\partial\mathrm{P_{EOS}}}{\partial\mathrm{T}}\right)_\rho\nabla\cdot\mathbf{u}, \quad (15)
$$

where $\lambda$ is the thermal conductivity and $c_v$ is the specific heat capacity at constant volume. Following the work of Fei et al.[50], we use the finite difference method to solve the above temperature equation, and the time discretization is realized using the fourth-order Runge–Kutta scheme:

$$
\mathrm{T}^{t+\Delta t} = \mathrm{T}^t + \frac{\Delta t}{6}\left(h_1 + 2h_2 + 2h_3 + h_4\right), h_1 = \mathrm{K}(\mathrm{T}^t), h_2
$$
$$
= \mathrm{K}\left(\mathrm{T}^t + \frac{\Delta t}{2}h_1\right), h_3 = \mathrm{K}\left(\mathrm{T}^t + \frac{\Delta t}{2}h_2\right), h_4 = \mathrm{K}(\mathrm{T}^t + \Delta t h_3), \quad (16)
$$

where K(T) denotes the right hand of Eq. (15). The coupling of the temperature field and the liquid-vapor phase change is achieved through the EOS of the fluid (Eq. (13)).

**Electrohydrodynamic multiphase flow**

The multiphase electrohydrodynamic model proposed in our recent work[39] is adopted, where the governing equation of the field strength follows Gauss's law:

$$
\nabla\cdot(\varepsilon\nabla\psi) = -\rho_e. \quad (17)
$$

The electric field strength is calculated as:

$$
\mathbf{E} = -\nabla\psi. \quad (18)
$$

Similar to our previous study[39], we consider a non-zero bulk charge model. The governing equation for the charge density evolution can be described as the following charge conservation equation[22,51]:

$$
\frac{\partial\rho_e}{\partial t} + \nabla\cdot(\rho_e\boldsymbol{u}) - \nabla\cdot(\sigma\nabla\psi) - \alpha\nabla^2\rho_e = 0. \quad (19)
$$

From the left to the right of the above equation, the first term stands for the charge relaxation, the second term accounts for charge convection, the third term stands for the Ohmic conduction, and the last term represents the charge diffusion. Usually, the charge diffusion can be ignored when charge relaxation and charge convection are dominated. By introducing characteristic variables $l_{ch} = l_r$, $u_{ch} = \varepsilon_l l_r E_0^2/\mu_l$, $t_{ch} = t_{diff}$, $\rho_{ech} = \varepsilon_l E_0$, $\sigma_{ch} = \sigma_l$ and $\alpha_{ch} = \varepsilon_l l_r^2 E_0^2/\mu_l$, the above charge conservation equation can be normalized as:

$$
\frac{t_e}{t_{diff}}\frac{\partial\rho_e^*}{\partial t^*} + \frac{t_e}{t_{conv}}\nabla\left(\rho_e^*\boldsymbol{u}^*\right) - \nabla(\sigma^*\nabla\psi^*) - \alpha^*\nabla^2\rho_e^* = 0. \quad (20)
$$

In many previous numerical studies for EHD multiphase flows, charge relaxation and charge convection were neglected when $t_e \ll t_{conv}$ and $t_e \ll t_{diff}$, simplifying Eq. (19) to $\nabla\cdot(\sigma\nabla\psi) = 0$. However, recent studies have highlighted the necessity of considering these effects explicitly. For instance, both Sengupta[52] and Wagoner et al.[53] reported that the jetting of sub-droplets only occurs under finite electric Reynolds numbers $Re_e$. When $Re_e$ approaches 0, the droplet exhibits an end-pinching state with conical ends. The D3Q19 non-orthogonal multiple-relaxation-time collision model is utilized to solving the charge conservation equation (Eq. 19):

$$
\mathbf{m}_{\rho_e}^* = \mathbf{M}f_{\rho_e,i}^* = \left(\mathbf{I} - \mathbf{S}_{\rho_e}\right)\mathbf{m}_{\rho_e} + \mathbf{S}_{\rho_e}\mathbf{m}_{\rho_e}^{eq} + \Delta t\left(\mathbf{I} - \frac{\mathbf{S}_{\rho_e}}{2}\right)\boldsymbol{R}_{\rho_e}
$$
$$
+ \Delta t\boldsymbol{C}_{\rho_e} + 0.5\Delta t^2\partial_t\left(\boldsymbol{C}_{\rho_e}\right). \quad (21)
$$

Considering the necessity of internal iterations at each time step in solving the Poisson equation, to save the computational cost, the D3Q7 single-relaxation-time collision operator is adopted in solving the Poisson equation (Eq. 17):

$$f^*_{\psi,i} = f_{\psi,i} - \frac{1}{\tau_\psi}\left(f_{\psi,i} - f^{eq}_{\psi,i}\right) + \Delta t' C_{\psi,i} + 0.5\Delta t'^2 \partial_t\left(C_{\psi,i}\right), \quad (22)$$

where $\Delta t' = 1$ is the inner iteration time step. The details of above multiphase EHD model and the explicit expression for $\mathbf{m}^{eq}_{\rho_e}, \mathbf{R}_{\rho_e}, \mathbf{C}_{\rho_e}, f^{eq}_{\psi,i}$ and $C_{\psi,i}$ can be found in our previous work[39]. According to CE analysis, the above LBM model can reproduce the charge conservation equation (Eq. 19) and Poisson equation (Eq. 17) for electric field, the corresponding macroscopic variables can be expressed as:

$$\rho_e = \sum_i f_{\rho_e,i},$$
$$\psi = \frac{\sum_{i=1}^6 f_{\psi,i}}{1 - \tilde{\omega}\left(|e_0|^2\right)}, \quad (23)$$

where $\tilde{\omega}$ stands for the wights of D3Q7 model. The coupling of the fluid flow and electric field is achieved by the leaky dielectric model as introduced in Eq. (1) without electrostriction force term.

## Model validations

The aforementioned phase-change multiphase flow model and EHD multiphase flow model have been extensively validated in our previous works[39,50,54]. Besides, three sets of additional validation cases are provided in supplementary information, i.e. evaporation of stationary droplets (Supplementary Fig. 2), droplets deformation in an electric field (Supplementary Fig. 3), and stretching of bubbles in an electric field (Supplementary Fig. 4). Through comparison of current simulation results with analytical solutions, previous simulation results, and experimental data, the accuracy of employed models are comprehensively validated.

## Data availability

The data supporting the findings of this study are available from the corresponding authors upon reasonable request.

## Code availability

All numerical codes used in this paper are available from the corresponding authors upon reasonable request.

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

## Acknowledgements

Support from National Key R&D Program of China (Grant No. 2022YFF0503501), the UK Engineering and Physical Sciences Research Council under the project "UK Consortium on Mesoscale Engineering Sciences (UKCOMES)" (Grant No. EP/X035875/1) and Strategic Priority Research Program of the Chinese Academy of Sciences (XDB0910102) are gratefully acknowledged. This work made use of computational support by CoSeC, the Computational Science Centre for Research Communities, through UKCOMES.

## Author contributions

G.W. and K.H.L. conceptualized the project; G.W. performed the simulations and analyzed the simulation data, supported by J.Y., T.L., L.F. and X.Z.; J.Z., K.L. and K.H.L. supervised the research. All authors contributed to interpreting the results of the study and editing the paper.

## Competing interests

The authors declare no competing interests.
