## [Transparent Peer Review file · Communications Physics]

Mesoscopic insights into effects of electric field on pool boiling for leaky dielectric fluids

Corresponding Author: Professor Kai Luo

Version 0:

Reviewer comments:

Reviewer #1

(Remarks to the Author)

In this paper, the authors developed a novel mesoscopic phase-change electrohydrodynamic (EHD) lattice Boltzmann method (LBM) model. Then, a large number of simulations with influencing parameters were studied, including electric field strength, residual conductivity, etc., and the effects of these parameters on heat transfer enhancement were obtained and the mechanism was analyzed. The topic of this study involves phase change and multi-physics coupling, which is very challenging. However, this paper adopts the most cutting-edge numerical methods and obtains a lot of quantitative data, which provides a solid support for the mechanism of how the electric field affects boiling heat transfer. It is recommended to publish with a minor revision.

There are a few suggestions:

- (1) An innovative point in the numerical aspect of this paper is "an electrohydrodynamic model incorporating charge convection and relaxation". Actually, this model has been discussed in many works in recent years, but this paper does not introduce the relevant literature. Another question: does the introduction of this model have an significant influence on the main conclusions of this paper, especially the overall heat transfer enhancement effect?
- (2) The organization of the paper is suggested to improve. 1)The initial results are with the water–vapor system, which is not a leaky dielectric fluid. 2) the results of the single bubble may be introduced before the results with large number of bubbles?
- (3) This paper mentions some theoretical model expansions, mainly the fitting of parameters, and the basis for introducing adjusting parameters and the scope of application are not fully discussed.

Minor points:

In Table 1, change "Relative permittivity" to "permittivity"

Reviewer #2

(Remarks to the Author)

The manuscript presents a numerical study of pool boiling under a uniform electric field of a leaky dielectric fluid (water) using a mesoscopic lattice Boltzmann method. The aim is to investigate the boiling of leaky dielectric fluids, which is poorly studied in the literature due to the significant enhancement effects of the electric forces. The work is novel, clearly written, and effective in highlighting several effects of the electric field that are difficult to investigate only experimentally. The paper can be considered for publication after some points have been addressed:

- 1) Picture labels and references of the supplementary materials do not comply with the text; please revise.
- 2) Lines 64-67: a classification based on electrical conductivity is given to define conductive, insulating and leaky dielectric fluids. Please comment on how the conductivity values were selected.
- 3) Line 110: Is T_c the critical temperature? Specify it in the text.
- 4) In Table 1 the relative permittivity is given as the product of the vacuum permittivity and the dimensionless relative permittivity. It appears that a value of 80 has been chosen for the relative permittivity of water, which is the typical value at ambient temperature. However, for the temperature considered, it should be around 22 (source: NIST), resulting in a value of about 1.95×10^{-10} F/m. Please comment on this.
- 5) Lines 133-134: the timescales are dimensional (seconds).
- 6) Please discuss on the choice of the mesh resolution (line 136).
- 7) Lines 147-150: a recent work measured the bubbles distribution and the dry area of pool boiling with electric field:

<https://doi.org/10.1016/j.applthermaleng.2025.125919>. There are some experimental pictures that corroborate also your discussion related to figure 4.

8) Equation 2 fits well with the results in the paper; has this been presented in other work or is it a new development? Please cite the reference or add an explanation of its derivation.

9) Figure 1-c: The caption reads "droplets" but there are "bubbles".

10) It seems that equation 3 provides a dimensional heat flux (W/m²) while in the figures it is considered as non dimensional. Please rectify.

11) Limitations of hydrodynamic theories should be cited.

12) It is not so clear where the charge density is evaluated in equation 6 (at which z? Close to the heated wall?).

13) Are the forces shown in figure 5 Coulomb forces?

Reviewer #3

(Remarks to the Author)

The paper presents the mesoscopic insights into effects of electric field on pool boiling for leaky dielectric fluids. The paper is well written and easy to read. The author presented significant results and citations are adequate. I would recommend to publish this paper in present form.

Version 1:

Reviewer comments:

Reviewer #1

(Remarks to the Author)

The authors have answered all my comments, and I can accept its publication.

Reviewer #2

(Remarks to the Author)

The authors addressed all the points and the paper can be considered for publication.

Journal: Communications Physics

Accession Code: COMMSPHYS-25-0019

Title: Mesoscopic insights into effects of electric field on pool boiling for leaky dielectric fluids.

Authors: Geng Wang, Junyu Yang, Timan Lei, Linlin Fei, Xiao Zhao, Jianfu Zhao, Kai Li, Kai H. Luo

Response to Referee #1

General comments: In this paper, the authors developed a novel mesoscopic phase-change electrohydrodynamic (EHD) lattice Boltzmann method (LBM) model. Then, a large number of simulations with influencing parameters were studied, including electric field strength, residual conductivity, etc., and the effects of these parameters on heat transfer enhancement were obtained and the mechanism was analyzed. The topic of this study involves phase change and multi-physics coupling, which is very challenging. However, this paper adopts the most cutting-edge numerical methods and obtains a lot of quantitative data, which provides a solid support for the mechanism of how the electric field affects boiling heat transfer. It is recommended to publish with a minor revision. There are a few suggestions:

Reply:

We are very grateful for the reviewer's recommendation to publish our paper with a minor revision. Following the reviewer's helpful comments, we have revised the manuscript accordingly. The changes are highlighted in red in the revised manuscript. Our responses to the reviewer's specific questions are given below.

1. An innovative point in the numerical aspect of this paper is "an electrohydrodynamic model incorporating charge convection and relaxation". Actually, this model has been discussed in many works in recent years, but this paper does not introduce the relevant literature. Another question: does the introduction of this model have a significant influence on the main conclusions of this paper, especially the overall heat transfer enhancement effect?

Reply:

We sincerely thank the reviewer for the constructive comments and suggestions. In response, we have now included the relevant literature. In particular, we highlight studies

explicitly comparing models incorporating finite electric Reynolds numbers (i.e., considering charge relaxation and convection) with models neglecting charge transport effects. Such as *“However, recent studies have highlighted the necessity of considering these effects explicitly. For instance, both Sengupta⁵¹ and Wagoner et al.⁵² reported that the jetting of sub-droplets only occurs under finite electric Reynolds numbers Re_e . When Re_e approaches 0, the droplet exhibits an end-pinching state with conical ends.”*

The importance of the charge convection and relaxation depends on the relevant time scales. Within our investigated parameter range, charge relaxation and convection effects are non-negligible. To clarify this, we have added the following statement in the revised manuscript: *“It should be pointed out that, the charge relaxation time $t_e = \epsilon_l/\sigma_l \sim 10^{-2} - 10^1$ s is longer than both the charge diffusion time $t_{diff} = \mu_l l_r/\gamma \sim 10^{-4}$ s and the charge convection time $t_{conv} = \mu_l/(\epsilon_l E_0^2) \sim 10^{-2}$ s, according to the non-dimensional analysis in the Method section, the charge relaxation (related to $t_e/t_{diff} = 10^2 - 10^5$) and charge convection (related to electric Reynolds numbers $Re_e = t_e/t_{conv} = 10^0 - 10^3$) should be included.”* And *“By introducing characteristic variables $l_{ch} = l_0$, $u_{ch} = \epsilon_l l_0 E_0^2/\mu_l$, $t_{ch} = t_{diff}$, $\rho_{ech} = \epsilon_l E_0$, $\sigma_{ch} = \sigma_l$ and $\alpha_{ch} = \epsilon_l l_0^2 E_0^2/\mu_l$, the above charge conservation equation can be normalized as:*

$$\frac{t_e}{t_{diff}} \frac{\partial \rho_e^*}{\partial t^*} + \frac{t_e}{t_{conv}} \nabla(\rho_e^* \mathbf{u}^*) - \nabla(\sigma^* \nabla \psi^*) - \alpha^* \nabla^2 \rho_e^* = 0, \quad (20)$$

In many previous numerical studies for EHD multiphase flows, charge relaxation and charge convection were neglected when $t_e \ll t_{conv}$ and $t_e \ll t_{diff}$, simplifying Eq. (19) to $\nabla \cdot (\sigma \nabla \psi) = 0$. However, recent studies have highlighted the necessity of considering these effects explicitly. For instance,...” Therefore, in the limit where charge relaxation and charge convection are neglected ($t_e = 0$, i.e., $T_\sigma = 0$), the findings presented in the Effect of Electrical Conductivity section would no longer hold.

2. The organization of the paper is suggested to improve. 1)The initial results are with the water–vapor system, which is not a leaky dielectric fluid. 2) the results of the single bubble may be introduced before the results with large number of bubbles?

Reply:

We sincerely thank the reviewer for these valuable suggestions.

(1) We apologise for the confusion caused by our initial description. The simulated fluids of initial results are based on the hydrodynamic of the water–vapor system, while the electrical

properties are chosen as leaky dielectric to clarify the effects of charge relaxation on EF pool boiling. To avoid confusion, we have clarified this point explicitly in the manuscript by adding: *“it should be noted that the electrical properties used in this study differ from those of realistic water–vapor systems. Higher liquid electrical permittivity and lower electrical conductivity were employed to highlight the effects of charge relaxation.”* and *“On the other hand, in a realistic water–vapor system, t_e is several orders of magnitude lower than in our simulations, resulting in negligible charge convection and charge relaxation effects. To fully incorporate the leaky dielectric effects (charge convection and charge relaxation), higher liquid electrical permittivity and lower electrical conductivity were employed in our simulations.”*.

(2) The purpose of the single-bubble simulations is primarily to explain the underlying bubble dynamics observed in large-scale EF pool boiling, such as the enhanced bubble departure frequency and reduced bubble size. Therefore, we have placed this section later in the manuscript to maintain a logical flow of presentation. *“To clarify the mechanisms of enhanced bubble departure observed in pool boiling, single-bubble simulations are introduced to elucidate fundamental bubble dynamics under electric fields.”*

3 This paper mentions some theoretical model expansions, mainly the fitting of parameters, and the basis for introducing adjusting parameters and the scope of application are not fully discussed.”

Reply:

We sincerely thank the reviewer for this insightful comment, which has helped us to further clarify the theoretical basis and robustness of our model significantly. Regarding the free parameter $a = 0.98$ in Equation (2), we have explained in our revised manuscript by adding *“The fitting parameter $a = 0.98$ is selected based on Berghmans’s³ original model constant $a = 1.0$, with a slight adjustment to ensure accurate prediction of bubble sizes under electrohydrodynamic conditions.”*

In Equation (5), two distinct sets of fitting parameters were selected separately for conventional pool boiling and EF pool boiling, reflecting the different driving mechanisms between these scenarios. Conventional pool boiling is primarily driven by buoyancy and surface tension, whereas EF pool boiling introduces additional complex mechanisms arising from electric field. Thus, the adjustment in fitting parameters accounts for the altered bubble dynamics unique to EF pool boiling. Both sets of parameters demonstrated robust predictive capability across a wide spectrum of heat flux, as illustrated in Fig. 2(c) and Fig. 5(d).

Specifically, for EF pool boiling, the fitting parameters accurately predicted the heat flux over broad ranges of electric field strengths and conductivities. We have added the following clarification to the revised manuscript: "*As illustrated in the figures, the fitting parameters result in high prediction accuracy within the investigated parameters.*". On the other hand, we acknowledge that the current fitting parameters have limitations, as their validity has only been explicitly verified within the Ja range of the current work. Further experimental and numerical studies beyond the current parameters are required, and we intend to address this in future research. To emphasise this point clearly, we have added the following sentence to the revised manuscript: "*Consequently, future experimental and numerical investigations focusing on realistic leaky dielectric fluids across a wider range of operating parameters are essential. Such studies would further enhance our understanding of electric field effects on pool boiling phenomena and validate the applicability and robustness of our proposed theoretical model for heat flux.*"

4. Minor points: In Table 1, change “Relative permittivity” to "permittivity"

Reply:

We have corrected this error in Table 1.

Journal: Communications Physics

Accession Code: COMMSPHYS-25-0019

Title: Mesoscopic insights into effects of electric field on pool boiling for leaky dielectric fluids.

Authors: Geng Wang, Junyu Yang, Timan Lei, Linlin Fei, Xiao Zhao, Jianfu Zhao, Kai Li, Kai H. Luo

Response to Referee #2

General comments: The manuscript presents a numerical study of pool boiling under a uniform electric field of a leaky dielectric fluid (water) using a mesoscopic lattice Boltzmann method. The aim is to investigate the boiling of leaky dielectric fluids, which is poorly studied in the literature due to the significant enhancement effects of the electric forces. The work is novel, clearly written, and effective in highlighting several effects of the electric field that are difficult to investigate only experimentally. The paper can be considered for publication after some points have been addressed:

Reply:

We sincerely thank the reviewer for the highly positive comments and the recommendation for the paper's acceptance, in principle. According to the reviewer's suggestions, we have carefully revised the manuscript, with the changes highlighted in red in our manuscript. Below are our responses to the reviewer's specific questions.

1. Picture labels and references of the supplementary materials do not comply with the text; please revise.

Reply:

Thank the reviewer very much for pointing out the mix-up. We have carefully revised all picture labels and references to ensure they are fully consistent with the main text and supplementary materials

2. Lines 64-67: a classification based on electrical conductivity is given to define conductive, insulating and leaky dielectric fluids. Please comment on how the conductivity values were selected.

Reply:

The conductivity values were selected based on criteria from the literature (Saville, 1997; Melcher & Taylor, 1969). Specifically, fluids with conductivity in the range of $10^{-5} \sim 10^{-12}$ S/m are typically classified as leaky dielectric fluids, balancing charge conduction and charge relaxation effects. Fluids with higher conductivity values are generally considered conductive fluids, whereas those with lower conductivity values are typically classified as insulating fluids. We have added this clarification and relevant references to the revised manuscript.

3. Line 110: Is T_c the critical temperature? Specify it in the text.

Reply:

We apologise for the ambiguity and have clarified this in the revised manuscript by adding " *T_c denotes the critical temperature of the fluid.*"

4. In Table 1 the relative permittivity is given as the product of the vacuum permittivity and the dimensionless relative permittivity. It appears that a value of 80 has been chosen for the relative permittivity of water, which is the typical value at ambient temperature. However, for the temperature considered, it should be around 22 (source: NIST), resulting in a value of about 1.95×10^{-10} F/m. Please comment on this.

Reply:

We sincerely appreciate the reviewer's insightful point. The fluid properties adopted in our simulations correspond to the density ratio of a typical water–vapour system, while the electrical properties are chosen to highlight the possible effects of conductivity and permittivity. Specifically, we employed lower electrical conductivity and higher permittivity values than realistic values.

However, this assumption does not affect the main conclusions of our study, as our primary aim is to elucidate the effects of electric fields on pool boiling phenomena for leaky dielectric fluids. To explicitly clarify this point, we have added the following sentence in our revised manuscript: "*it should be noted that the electrical properties used in this study differ from those of realistic water–vapor systems. Higher liquid electrical permittivity and lower electrical conductivity were employed to highlight the effects of charge relaxation.*" and "*On the other hand, in a realistic water–vapor system, t_e is several orders of magnitude lower than in our simulations, resulting in negligible charge convection and charge relaxation effects. To fully incorporate the leaky dielectric effects (charge convection and charge relaxation), higher liquid electrical permittivity and lower electrical conductivity were employed in our*

simulations.” and the theoretical analysis “It should be pointed out that, the charge relaxation time $t_e = \epsilon_l/\sigma_l \sim 10^{-2} - 10^1$ s is longer than both the charge diffusion time $t_{diff} = \mu_l l_r/\gamma \sim 10^{-4}$ s and the charge convection time $t_{conv} = \mu_l/(\epsilon_l E_0^2) \sim 10^{-2}$ s, according to the non-dimensional analysis in Method section, the charge relaxation (related to $t_e/t_{diff} = 10^2 - 10^5$) and charge convection (related to electric Reynolds numbers $Re_e = t_e/t_{conv} = 10^0 - 10^3$) should be included.” And “By introducing characteristic variables $l_{ch} = l_0$, $u_{ch} = \epsilon_l l_0 E_0^2/\mu_l$, $t_{ch} = t_{diff}$, $\rho_{ech} = \epsilon_l E_0$, $\sigma_{ch} = \sigma_l$ and $\alpha_{ch} = \epsilon_l l_0^2 E_0^2/\mu_l$, the above charge conservation equation can be normalized as:

$$\frac{t_e}{t_{diff}} \frac{\partial \rho_e^*}{\partial t^*} + \frac{t_e}{t_{conv}} \nabla(\rho_e^* \mathbf{u}^*) - \nabla(\sigma^* \nabla \psi^*) - \alpha^* \nabla^2 \rho_e^* = 0. \quad (20)$$

In many previous numerical studies for EHD multiphase flows, charge relaxation and charge convection were neglected when $t_e \ll t_{conv}$ and $t_e \ll t_{diff}$, simplifying Eq. (19) to $\nabla \cdot (\sigma \nabla \psi) = 0$. However, recent studies have highlighted the necessity of considering these effects explicitly. For instance, both Sengupta⁵² and Wagoner et al.⁵³ reported that the jetting of sub-droplets only occurs under finite electric Reynolds numbers Re_e . When Re_e approaches 0, the droplet exhibits an end-pinching state with conical ends.”

Furthermore, we acknowledge the limitations inherent in our current approach and have highlighted the importance of adopting realistic water–vapour systems in future studies. To address this clearly, we have added the following statement in the discussion section: "Consequently, future experimental and numerical investigations focusing on realistic leaky dielectric fluids across a wider range of operating parameters are essential. Such studies would further enhance our understanding of electric field effects on pool boiling phenomena and validate the applicability and robustness of our proposed theoretical model for heat flux."

5. Lines 133-134: the timescales are dimensional (seconds).

Reply:

We thank the reviewer for pointing this out. The original values were shown in lattice units. To maintain consistency, we have converted them into seconds.

6. Please discuss on the choice of the mesh resolution (line 136).

Reply:

The chosen mesh resolution ($dx = 100 \mu\text{m}$) is chosen to ensure our domain size $L_x/L_y/L_z \gg l_r \gg dx$ (where $L_x = 80\text{mm}$, $L_y = 80\text{mm}$, $L_z = 40\text{mm}$ are the computational

domain, $l_r = 1.9\text{mm}$ is the bubble length scale) are satisfied. Therefore, it is reasonable to conclude that the current mesh resolution can resolve a significant number of bubbles (up to hundreds), providing sufficient accuracy for capturing essential bubble dynamics. To clarify this explicitly, we have included the following sentence in our revised manuscript: *“It should be noted that the condition $L_{x,y,z} \gg l_r \gg dx$ is well satisfied with the current mesh resolution, indicating that the computational domain is sufficiently large relative to both the bubble length scale and the grid spacing. Consequently, the present mesh resolution can resolve a substantial number of bubbles (up to several hundred).”*

7. Lines 147-150: a recent work measured the bubbles distribution and the dry area of pool boiling with electric field: <https://doi.org/10.1016/j.applthermaleng.2025.125919>. There are some experimental pictures that corroborate also your discussion related to figure 4.

Reply:

We sincerely thank the reviewer for bringing this new and highly relevant reference to our attention. We have carefully reviewed this paper, and the findings provided valuable insights and strong experimental corroboration for our numerical simulations. To highlight this, we have added the following clarification in our revised manuscript: *“Recently, Graffiedi et al.⁴⁷ employed high-resolution optical diagnostics to investigate EF pool boiling of the dielectric fluid. Their study similarly observed reductions in continuous bubble footprints and increases in bubble departure frequency with applied electric fields. These experimental results align closely with our simulation findings, providing strong support for the above analysis regarding bubble nucleation sites.”*

8. Equation 2 fits well with the results in the paper; has this been presented in other work or is it a new development? Please cite the reference or add an explanation of its derivation.

Reply:

Equation (2) is an extension of the traditional Berghmans’s hydrodynamic model for bubble size. Compared to the original Berghmans’s model, we introduced an adjustment function $f(T_\sigma)$ to explicitly account for the influence of electrical conductivity. We have added *“by extending Berghmans’s hydrodynamic model⁴⁰, ...”* to the revised manuscript.

9. Figure 1(c): The caption reads "droplets", but there are "bubbles".

Reply:

Thank the reviewer very much for noticing our oversight. We have corrected the caption in Figure 1(c).

10. It seems that equation 3 provides a dimensional heat flux (W/m²) while in the figures it is considered as non dimensional. Please rectify.

Reply:

We sincerely apologies for any confusion caused by our previous description. The heat flux values presented in the original figures were shown in lattice units. To maintain consistency, we have uniformly non-dimensionalised the heat flux using Zuber's correlation ($q_r = h_{fg}\rho_g\sqrt{g\gamma(\rho_l - \rho_g)}$) in our revised manuscript and updated all relevant figures accordingly. This correction is explicitly clarified in the manuscript by adding the following statement: "*The evolution of non-dimensionalised transient heat flux $\tilde{q} = Q_h/q_r$ is plotted in Fig. 1(e), the term $q_r = h_{fg}\rho_g\sqrt{g\gamma(\rho_l - \rho_g)}$ stands for the Zuber's correlation⁴³.*"

11. Limitations of hydrodynamic theories should be cited.

Reply:

We sincerely thank the reviewer for this valuable suggestion. The original Berghmans hydrodynamic theory assumes flat heating surfaces; thus, it has not been validated for complex geometries. To clarify this limitation explicitly, we have added the following statement to our revised manuscript: "*It should be noted that the above hydrodynamic theoretical model is derived based on flat plate assumption and has not been validated for complex geometrical heating surfaces.*"

Additionally, the extended hydrodynamic model proposed in our study is applicable specifically to leaky dielectric and insulating fluids. To emphasize it, we have included the following clarification: "*It should be pointed out that the proposed adjustment function is only validated for leaky dielectric and insulating fluids. For purely insulating fluids (with infinite T_σ), Eq. (2) reduces to the original Berghmans's hydrodynamic model⁴⁰.*"

12. It is not clear where the charge density is evaluated in Equation (6) (at which z? Close to the heated wall?).

Reply:

The charge density in Equation (6) was evaluated near the heated wall surface, as the charge accumulation effects are strongest in this region. We have clarified this by adding "*we*

examined the accumulated charge density $\sum \bar{\rho}_{e,z}$ close to the superheated surface (where $dH = (Z - H_w)/L_z$), where the charge accumulation effects are strongest.” in the revised manuscript.

13. Are the forces shown in Figure 5 Coulomb forces?

Reply:

The forces depicted in Figure 5 represent the combined effects of Coulomb and polarization forces. To clarify this point, we have added the following statement in our revised manuscript: "*As illustrated in Figure 4(a), the electric force (comprising both Coulomb and polarization contributions) acting on growing bubbles exhibits a significant size dependence.*"

Journal: Communications Physics

Accession Code: COMMSPHYS-25-0019

Title: Mesoscopic insights into effects of electric field on pool boiling for leaky dielectric fluids.

Authors: Geng Wang, Junyu Yang, Timan Lei, Linlin Fei, Xiao Zhao, Jianfu Zhao, Kai Li, Kai H. Luo

Response to Referee #3

General comments: *The paper presents the mesoscopic insights into effects of electric field on pool boiling for leaky dielectric fluids. The paper is well written and easy to read. The author presented significant results and citations are adequate. I would recommend to publish this paper in present form.*

Reply: We are very grateful to the reviewer for recommendation to publish our manuscript.

Journal: Communications Physics

Accession Code: COMMSPHYS-25-0019

Title: Mesoscopic insights into effects of electric field on pool boiling for leaky dielectric fluids.

Authors: Geng Wang, Junyu Yang, Timan Lei, Linlin Fei, Xiao Zhao, Jianfu Zhao, Kai Li, Kai H. Luo

Response to Referee #1

General comments: *The authors have answered all my comments, and I can accept its publication.*

Reply: We are very grateful to the reviewer for recommendation to publish our manuscript.

Journal: Communications Physics

Accession Code: COMMSPHYS-25-0019

Title: Mesoscopic insights into effects of electric field on pool boiling for leaky dielectric fluids.

Authors: Geng Wang, Junyu Yang, Timan Lei, Linlin Fei, Xiao Zhao, Jianfu Zhao, Kai Li, Kai H. Luo

Response to Referee #2

General comments: *The authors addressed all the points and the paper can be considered for publication.*

Reply: We are very grateful to the reviewer for recommendation to publish our manuscript.